# Mutual Effects of Orexin and Bone Morphogenetic Proteins on Gonadotropin Expression by Mouse Gonadotrope Cells

**DOI:** 10.3390/ijms23179782

**Published:** 2022-08-29

**Authors:** Yoshiaki Soejima, Nahoko Iwata, Nanako Nakayama, Shinichi Hirata, Yasuhiro Nakano, Koichiro Yamamoto, Atsuhito Suyama, Kohei Oguni, Takahiro Nada, Satoshi Fujisawa, Fumio Otsuka

**Affiliations:** 1Department of General Medicine, Graduate School of Medicine, Dentistry and Pharmaceutical Sciences, Okayama University, 2-5-1 Shikata-cho, Kitaku, Okayama 700-8558, Japan; 2Department of Nephrology, Rheumatology, Endocrinology and Metabolism, Graduate School of Medicine, Dentistry and Pharmaceutical Sciences, Okayama University, 2-5-1 Shikata-cho, Kitaku, Okayama 700-8558, Japan

**Keywords:** bone morphogenetic protein (BMP), clock, gonadotropin, orexin, pituitary

## Abstract

Orexin plays a key role in the regulation of sleep and wakefulness and in feeding behavior in the central nervous system, but its receptors are expressed in various peripheral tissues including endocrine tissues. In the present study, we elucidated the effects of orexin on pituitary gonadotropin regulation by focusing on the functional involvement of bone morphogenetic proteins (BMPs) and clock genes using mouse gonadotrope LβT2 cells that express orexin type 1 (OX1R) and type 2 (OX2R) receptors. Treatments with orexin A enhanced LHβ and FSHβ mRNA expression in a dose-dependent manner in the absence of GnRH, whereas orexin A in turn suppressed GnRH-induced gonadotropin expression in LβT2 cells. Orexin A downregulated GnRH receptor expression, while GnRH enhanced OX1R and OX2R mRNA expression. Treatments with orexin A as well as GnRH increased the mRNA levels of Bmal1 and Clock, which are oscillational regulators for gonadotropin expression. Of note, treatments with BMP-6 and -15 enhanced OX1R and OX2R mRNA expression with upregulation of clock gene expression. On the other hand, orexin A enhanced BMP receptor signaling of Smad1/5/9 phosphorylation through upregulation of ALK-2/BMPRII among the BMP receptors expressed in LβT2 cells. Collectively, the results indicate that orexin regulates gonadotropin expression via clock gene expression by mutually interacting with GnRH action and the pituitary BMP system in gonadotrope cells.

## 1. Introduction

Orexins are neuropeptides that play important roles in various systems including the regulation of wakefulness, feeding behavior, emotion, and autonomic function [1]. Orexins are composed of two isoforms, orexin A (ORX; also known as hypocretin 1) and orexin B (hypocretin 2), both of which are derived from a common precursor peptide. Orexin A binds to orexin receptor type 1 (OX1R) and type 2 (OX2R), while orexin B binds selectively to OX2R [1,2]. Alongside its effects on the central nervous system, orexin has been reported to have major functions in peripheral tissues including the endocrine system [3,4].

Effects of orexins on gonadotrope cells have been shown in experiments using anterior pituitary cells and proestrus rodents, suggesting that both orexin isoforms can induce gonadotropin secretion via the orexin receptors [5]. Interestingly, in orexin-deficient narcoleptic humans, pulsatile LH secretion was found to be blunted, indicating that endogenous orexins are functionally involved in the regulation of the hypothalamo–pituitary–gonadal (HPG) axis in humans [6]. However, the molecular mechanism by which orexins activate the secretory activity of gonadotropins by gonadotrope cells has yet to be clarified.

The circadian system of mammals is composed of a central pacemaker in the suprachiasmatic nuculei (SCN) and subsidiary circadian clocks in nearly every body cell [7,8]. The circadian oscillators function throughout the body, including the endocrine organs [9]. At molecular levels, the circadian clock forms an autoregulatory negative-feedback transcriptional network [10]. Among the transcriptional activators, Clock and Bmal1 proteins form a heterodimer and positively regulate the expression of the Period (Per1, Per2) and Cryptochrome (Cry1, Cry2) genes, and then Per and Cry gene products dimerize and translocate into the nucleus to interact with Clock and Bmal1 proteins, leading to repression of their own transcription [7]. We previously reported that among the canonical clock genes, Bmal1 and Clock functionally regulate LHβ mRNA expression under the influence of GnRH by interacting with bone morphogenetic proteins (BMPs) in mouse gonadotrope cells [11].

There has been growing evidence that BMPs, which are members of the transforming growth factor (TGF)-β superfamily and induce bone formation, play various functional roles in endocrine tissues [12,13]. Several BMP ligands have been shown to exert regulatory functions in gonadotropin production by gonadotrope cells [14,15,16,17]. In mouse gonadotrope cells, it was revealed that treatments with BMP-6, -7, and -15 enhanced transcriptional activity of the follicle-stimulating hormone (FSH) [16]. It was also revealed that BMP-6 modulates gonadotropin-releasing hormone (GnRH)-induced luteinizing hormone (LH) production by altering the responsiveness to somatostatin analogs [18]. It has also been shown that intracellular signaling of BMPs interacted with intracellular signaling of mitogen-activated protein kinase (MAPK) in the regulation of LH expression via clock gene expression in a phase- and GnRH-dependent manner [11]. However, the regulatory mechanism involving BMPs, orexins and circadian rhythm in gonadotropin synthesis is still unclear.

In the present study, we investigated the functional roles of orexin in the regulation of gonadotropin expression by focusing on the interaction of the BMP system and clock gene expression using mouse gonadotrope cells.

## 2. Results

In our previous study, we confirmed the expression of orexin type 1 (OX1R) and type 2 (OX2R) receptors in mouse gonadotrope LβT2 cells [19]. To evaluate the functional roles of orexin in LβT2 cells in the present study, changes in gonadotropin expression after orexin A (10 to 300 nM) stimulation were evaluated with quantitative PCR in LβT2 cells in the presence or absence of GnRH (10 nM) under serum-free conditions for 24 h. As shown in Figure 1A, LHβ mRNA levels were revealed to be significantly enhanced by orexin A treatment in a dose-dependent manner without GnRH, and 100 nM of orexin A showed the maximum effect. FSHβ mRNA levels were also increased by orexin A treatment, but the difference was not significant in comparison with the control levels. Of note, GnRH treatment increased LHβ and FSHβ mRNA levels; however, the elevated gonadotropin mRNA levels were suppressed by orexin A treatment dose-responsively with 100 to 300 nM of orexin A showing the maximum effect.

The mutual effects of orexin and GnRH responsiveness in relation to clock gene expression were further evaluated. Among the core clock genes expressed in gonadotrope cells [20], Bmal1 and Clock are considered to be functional regulators for LHβ mRNA expression by LβT2 cells as we earlier reported [11]. As shown in Figure 1B, it was revealed that orexin A (100 nM) treatment significantly suppressed GnRH receptor (GnRHR) mRNA expression, whereas GnRH (10 nM) increased OX1R and OX2R mRNA expression by LβT2 cells for 24 h in serum-free conditions. In addition, treatments with orexin A as well as GnRH were found to upregulate Bmal1 and Clock mRNA expression (Figure 1C).

The involvement of BMP signaling in orexin receptor signaling and clock gene expression was then examined. The BMP system has also been postulated to be a functional modulator of LH expression by LβT2 cells [11,18]. As shown in Figure 2A, treatments with BMP-6 and -15 (30 ng/mL) resulted in upregulation of both OX1R and OX2R mRNA expression in LβT2 cells. It was also found that BMP-6 significantly enhanced Bmal1 mRNA expression, while BMP-15 significantly enhanced Clock mRNA expression.

Of interest, it was found that treatment with orexin A (100 nM) augmented the Smad1/5/9 phosphorylation induced by stimulation with BMP-6 or BMP-15 (30 ng/mL) for 1 h (Figure 2B). To approach the underlying mechanism by which orexin augmented BMP signaling, the expression levels of key BMP receptors were examined. As a result, treatment with orexin A was found to upregulate the mRNA expression of ALK-2 among the BMP type 1 receptors and the mRNA expression of BMPRII among the BMP type 2 receptors (Figure 2C). These results indicated that orexin A enhanced the BMP actions that induce clock gene expression, leading to gonadotropin expression by LβT2 cells.

## 3. Discussion

In the present study, the functional interrelationship of orexin A, GnRH, BMP signaling, and clock gene expression was uncovered by using mouse gonadotrope LβT2 cells (Figure 3). It was revealed that orexin A enhanced gonadotropin expression via upregulation of Bmal1/Clock expression but suppressed GnRH-induced gonadotropin expression in vitro. Based on the results showing that orexin A suppressed GnRHR expression and GnRH increased OX1R/OX2R and Bmal1/Clock mRNA expression, there seems to be a feedback system between the orexin and GnRH signaling in gonadotrope cells. Moreover, orexin A enhanced BMP-Smad1/5/9 phosphorylation, leading to induction of the expression of OX1R/OX2R and clock genes. These findings suggest the existence of a functional network among orexin A, GnRH, clock gene, and BMP signaling for the regulation of gonadotropin expression by gonadotrope cells.

Recent studies have shown that orexin A regulates gonadotropin secretion in cooperation with GnRH in the anterior pituitary. For instance, Martynska et al. showed that orexin A increased LH secretion with a high dose of GnRH (1 μM) but inhibited GnRH-stimulated gonadotropin release at lower doses of GnRH (1 nM) in primary anterior pituitary cells isolated from immature female rats [21]. Russel et al. also demonstrated that orexin A, at a higher dose of 100 nM, inhibited GnRH (1 nM)-stimulated LH release in dispersed pituitaries from proestrus female rats in a dose-dependent manner [22]. In the present study, orexin A by itself accelerated gonadotropin expression but suppressed GnRH-stimulated gonadotropin expression, suggesting that orexin action is closely linked to GnRH activity. Based on our results showing that orexin A potently downregulated GnRHR expression and that GnRH in turn increased the expression of OX1R/OX2R by gonadotrope cells, orexin is likely to have modulatory effects on gonadotropin expression by altering GnRH responsiveness, leading to a feedback regulation of gonadotropin expression in the presence of GnRH.

In an in vivo study on LH secretion, it was shown that orexin A reduced LH synthesis and release in immature female rats [23]. It was also shown that orexin A inhibited LH release in ovarian steroid-unprimed ovariectomized (OVX) rats, whereas orexin A induced LH secretion in estoradiol benzoate (EB)- and progesterone-pretreated OVX rats [24], suggesting that gonadal steroids are key factors for the effects of orexin on LH synthesis in vivo. Functional roles of orexins, mainly orexin A, have also been demonstrated in the central nervous system [25,26]. Orexins suppressed the pulsatile secretion of LH in OVX female rats via β-endorphin and corticotropin-releasing factor (CRF) receptor type-2 expressed in the hypothalamus [27,28,29]. Moreover, orexin A was shown to be involved in the suppression of GnRH neuron activity [30] and LH surge generation in OVX rodents [31]. The effects of orexin A on the hypothalamus were site-specifically analyzed, and it was shown that the rostral preoptic area (rPOA) and the medial preoptic area (mPOA) are critical for the effects of orexin on LH secretion [32]. These results suggest that both gonadal steroids and hypothalamic GnRH are critical for the effects of orexin including direct control by orexin of gonadotropin secretion in gonadotrope cells and indirect control of gonadotropin secretion via the modulation of endogenous GnRH activity in vivo.

Clock genes are also involved in gonadotropin expression in pituitary gonadotrope cells. Several studies have shown that clock genes exhibit circadian oscillations in mouse gonadotrope cells [11,33], mouse pituitary [34], and human pituitary tissues [35]. As for their biological roles in gonadotropin function, several studies have shown that Clock-mutant and Bmal1-knockout female mice had disruption of estrous cycles and a lack of proestrus LH surge [34,36,37]. We previously reported that clock genes are involved in regulation of LHβ expression in a phase-dependent manner by mouse gonadotrope cells [11]. Those results and the results of the present study indicate that clock genes are likely to have functional roles in the formation of LH surge by collaborating with orexin and BMPs. In the present study, we examined the time point of 24 h stimulation, at which LHβ and clock gene mRNA expression was effectively induced by GnRH treatment [11]. Further studies at different timepoints with various reagents would be needed to clarify the functional roles of clock genes in the modulation of gonadotropin secretion.

It is known that orexins and BMP signaling collaboratively modulate hormonal secretion in various endocrine tissues. We previously reported that orexin A enhanced FSH-induced progesterone production by downregulating BMP signaling in rat granulosa cells [38]. As for pituitary functions, it was reported that orexin A inhibited prolactin production by suppressing endogenous BMP activity in rat lactotrope cells [39] and that orexin A enhanced pro-opiomelanocortin (POMC) transcription by upregulating corticotropin-releasing hormone receptor (CRHR) signaling and by downregulating BMP signaling in mouse corticotrope cells [19]. It was further shown that the pituitary BMP system is involved in the regulation of LH expression via clock gene expression in mouse gonadotrope cells [11]. Given that the phase-dependent changes of clock gene expression are functionally linked to the pulsatile secretion of LH [11,34,36,37], it is possible that orexins and clock genes are collaboratively involved in the formation of biologic secretory rhythm of LH by interacting with BMP signaling.

Collectively, the results of the present study indicate that orexin by itself activates gonadotropin expression but suppresses GnRH-induced gonadotropin expression. A mutual interrelationship between orexin action and BMP signaling in gonadotrope cells was uncovered. Hence, it is possible that the pituitary BMP system plays functional roles in the modulation of orexin effects and clock gene expression, leading to fine-tuning of gonadotropin expression by gonadotrope cells (Figure 3). Control of orexins, clock genes, and the endogenous BMP system in pituitary gonadotropes could be a possible strategy for maintenance of the secretory pattern of gonadotropins and fertile control by modulating gonadotropin secretion.

## 4. Materials and Methods

### 4.1. Experimental Reagents

Recombinant human BMP-6 and BMP-15 were purchased from R&D Systems Inc. (Minneapolis, MN, USA). Human orexin A was purchased from Wako Pure Chemical Industries, Ltd. (Osaka, Japan) and GnRH human acetate salt was purchased from Sigma-Aldrich Co. Ltd. (St. Louis, MO, USA). LβT2 cells derived from a mouse gonadotrope cell line were cultured in Dulbecco’s modified Eagle’s medium (DMEM) containing 10% fetal calf serum (FCS) supplemented with penicillin-streptomycin in 12-well plates under a 5% CO_2_ atmosphere at 37 °C.

### 4.2. Quantitative Real-Time PCR Analysis

LβT2 cells (3 × 10^5^ cells/mL) were treated with orexin A (10–300 nM), GnRH (10 nM), and BMP-6 and -15 (30 ng/mL) in serum-free DMEM for 24 h incubation after adjustment to a serum-free condition for 3 h. The timepoint of 24 h for mRNA collection was determined on the basis of the finding that LHβ and clock gene expression was effectively induced by GnRH stimulation [11]. Total cellular RNAs were extracted using TRI Reagent^®^ (Cosmo Bio Co., Ltd., Tokyo, Japan) and RNA concentrations were evaluated using a NanoDrop^TM^ One spectrophotometer (Thermo Fisher Scientific, Waltham, MA, USA). Primer pairs for PCR were selected from different exons to eliminate PCR products caused by contaminated chromosomal DNA. Primer pairs for LHβ, Bmal1 and Clock were prepared as we reported earlier [11,40]. Other primer pairs for PCR were chosen as follows: 405–424 and 552–571 for ribosomal protein L19 (RPL19), a housekeeping gene (from GenBank accession #NM_009078), 162–181 and 375–394 for FSHβ (NM_008045), 671–690 and 902–921 for GnRHR (NM_010323), 735–754 and 874–893 for OX1R (NM_001357258), 310–329 and 476–495 for OX2R (NM_198962), 1201–1220 and 1415–1434 for ALK-2 (NM_001355049), 210–229 and 368–387 for ALK-3 (NM_009758), 1047–1066 and 1216–1235 for ALK-6 (NM_007560), 1199–1218 and 1332–1351 for ActRII (NM_001278579), and 2561–2580 and 2718–2737 for BMPRII (NM_001204). An RT reaction was performed using ReverTra Ace^®^ (TOYOBO Co., Ltd., Osaka, Japan) and then quantitative PCR (qPCR) analysis was performed using the LightCycler^®^ Nano real-time PCR system (Roche Diagnostic Co., Tokyo, Japan). After optimization of the annealing conditions and amplification efficiency [38], the expression levels of target gene mRNAs were calculated by the method using the ∆ threshold cycle (Ct). The relative expression of each mRNA was evaluated by the ∆Ct method, in which ∆Ct was calculated by subtracting the Ct values of RPL19 from those of the target genes. Each mRNA level of the target gene, normalized by RPL19 mRNA, was expressed as 2^−(∆Ct)^. The results were shown as ratios of target gene mRNA to RPL19 mRNA.

### 4.3. Western Immunoblotting Analysis

LβT2 cells (3 × 10^5^ cells/mL) were pretreated with orexin A (100 nM) in serum-free DMEM for 24 h. After 1 h stimulation with BMP ligands (30 ng/mL) as we previously reported [18], the cells were solubilized in 100 μl RIPA lysis buffer (Upstate Biotechnology, Lake Placid, NY, USA) containing 1 mM Na_3_VO_4_, 1 mM NaF, 2% SDS, and 4% β-mercaptoethanol. The cell lysates were then subjected to SDS-PAGE/immunoblotting analysis using antibodies against phospho-Smad1/5/9 (pSmad1/5/9) and total-Smad1 (tSmad1; Cell Signaling Technology, Inc., Beverly, MA, USA). The integrated signal densities were analyzed with the C-DiGit^®^ Blot Scanner System (LI-COR Biosciences, Lincoln, NE, USA).

### 4.4. Statistics

Data were obtained from at least three independent experiments with triplicate samples. All of the results are shown as means ± SEM. EZR, version 1.40 (Saitama Medical Center, Jichi Medical University, Saitama, Japan), which is a graphical user interface for R (The R Foundation for Statistical Computing, Vienna, Austria), was utilized in all statistical analyses. It is modified from R commander, which was designed in order to add frequently used functions in biostatistics [41]. Statistical analysis was performed by ANOVA with Tukey–Kramer’s post hoc test or unpaired *t-*test. *p* values < 0.05 were accepted as statistically significant.

## Figures and Tables

**Figure 1 ijms-23-09782-f001:**
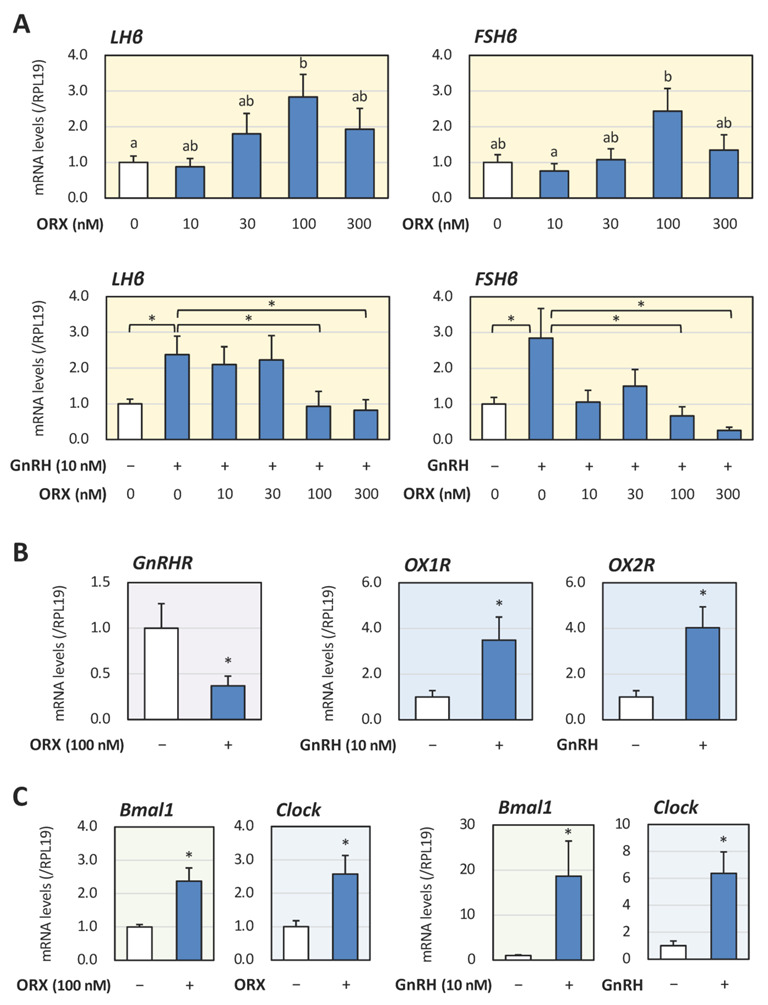
**Mutual effects of orexin A and GnRH on gonadotropin and clock gene expression by mouse gonadotrope cells.** (**A**) LβT2 cells (3 × 10^5^ cells/mL) were treated with the indicated concentration of orexin A (ORX) with or without GnRH (10 nM) in serum-free DMEM for 24 h. Total cellular RNAs were extracted and the mRNA levels of LHβ and FSHβ genes were standardized by RPL19 mRNA levels and expressed as fold changes. Results are shown as means ± SEM and were analyzed with ANOVA or the unpaired *t*-test. The values with different superscript letters are significantly different at *p* < 0.05; and * *p* < 0.05 between the indicated groups. (**B**,**C**) Cells (3 × 10^5^ cells/mL) were treated with ORX (100 nM) or GnRH (10 nM) in serum-free DMEM for 24 h. Total cellular RNAs were extracted and mRNA levels of the receptors for GnRH (GnRHR) and orexin (OX1R and OX2R; (**B**)) and clock genes (Bmal1 and Clock; (**C**)) were standardized by RPL19 levels and expressed as fold changes. Results are shown as means ± SEM and were analyzed with the unpaired *t*-test: * *p* < 0.05 between the indicated groups.

**Figure 2 ijms-23-09782-f002:**
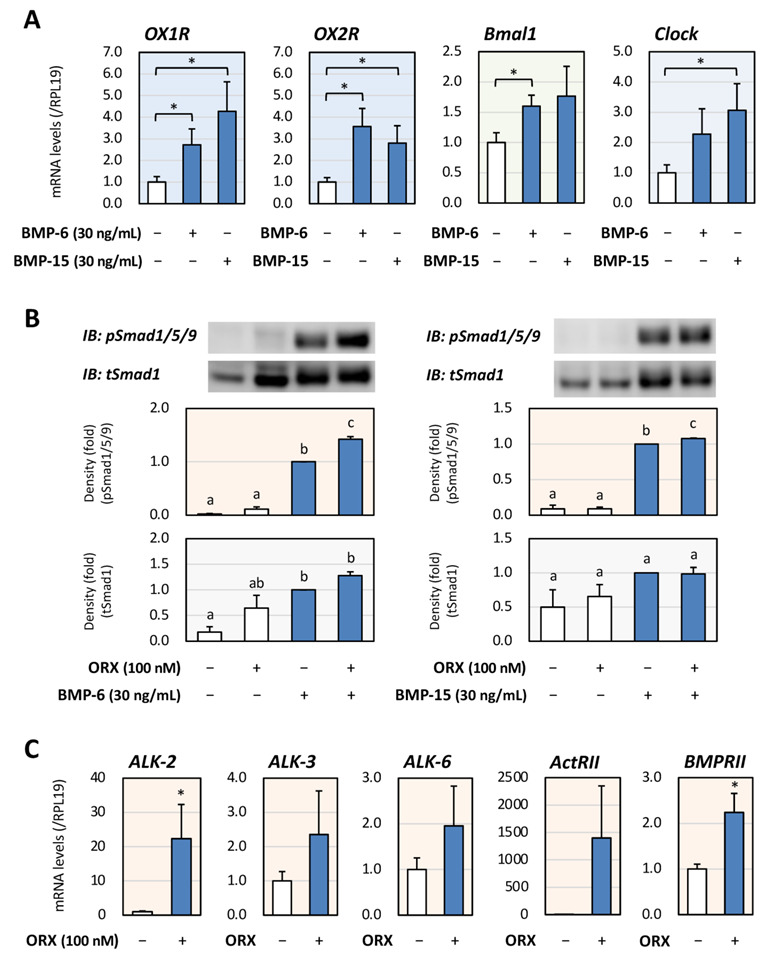
**Interaction of orexin A, BMP signaling and clock gene expression in gonadotrope cells.** (**A**) LβT2 cells (3 × 10^5^ cells/mL) were treated with BMP-6 and BMP-15 (each 30 ng/mL) in serum-free DMEM for 24 h. Total cellular RNAs were extracted and OX1R, OX2R, Bmal1, and Clock mRNA levels were standardized by RPL19 levels and expressed as fold changes. Results are shown as means ± SEM and were analyzed with the unpaired *t*-test: * *p* < 0.05 between the indicated groups. (**B**) Cells (3 × 10^5^ cells/mL) were pretreated with orexin A (ORX; 100 nM) in serum-free DMEM for 24 h. After 1 h stimulation with BMP-6 and BMP-15 (each 30 ng/mL), the cells were lysed and subjected to immunoblot (IB) analysis using antibodies that detect pSmad1/5/9 and tSmad1. The results are representative of those obtained from at least three independent experiments and were expressed as fold changes. Results are shown as means ± SEM and were analyzed with the unpaired *t*-test. The values with different superscript letters are significantly different at *p* < 0.05. (**C**) Cells (3 × 10^5^ cells/mL) were treated with ORX (100 nM) in serum-free DMEM for 24 h. Total cellular RNAs were extracted and the mRNA levels of BMP type 1 receptors (ALK-2, -3 and -6) and BMP type 2 receptors (ActRII and BMPRII) were standardized by RPL19 levels and expressed as fold changes. Results are shown as means ± SEM and were analyzed with the unpaired *t*-test: * *p* < 0.05 between the indicated groups.

**Figure 3 ijms-23-09782-f003:**
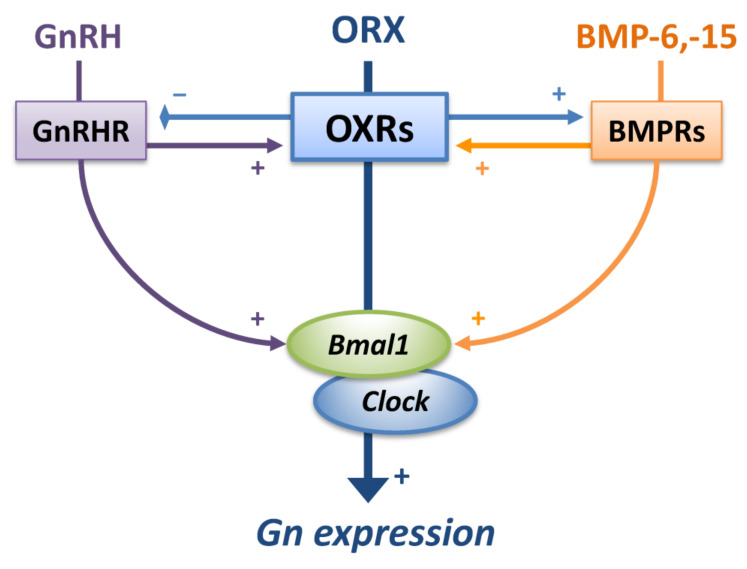
**Functional interaction of orexin A, BMP signaling and clock genes in gonadotrope cells.** Orexin A (ORX) stimulated the expression of gonadotropins (Gn) but suppressed the GnRH-induced expression of Gn. ORX reduced GnRHR mRNA levels, while GnRH increased OX1R/OX2R and Bmal1/Clock expression. ORX enhanced BMP signaling activity by upregulation of BMP receptors (BMPRs), while BMP-6 and -15 upregulated OX1R/OX2R and Bmal1/Clock mRNA expression. Thus, functionally mutual interactions among ORX, BMP signaling, and clock genes for gonadotropin regulation were shown in gonadotrope cells.

## Data Availability

Data is contained within the article.

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
