# Peer review of "Mutual Effects of Orexin and Bone Morphogenetic Proteins on Gonadotropin Expression by Mouse Gonadotrope Cells"

_ijms, 2022, doi:10.3390/ijms23179782_

Round 1

Reviewer 1 Report

In this study, Soejima et al. studied the role of orexins, important neuropeptides involved in wakefulness, feeding behavior, emotion, and autonomic function in regulation of gonadotropin expression. Using mouse gonadotrope LbT2 cells, the authors examined the expression of gonadotropins (LHb and FSHb), GnRH-receptor, orexin receptors (OX1R and OX2R) and clock genes (Bmal1 and Clock) by stimulation with orexin and/or GnRH. Then, the authors examined expression of orexin receptors and clock genes by BMP stimulation. The authors also examined effect of orexin stimulation on BMP-induced phosphorylation of smad proteins, and expression of BMP receptors by orexin stimulation. The authors concluded mutual interaction of orexin, GnHR, and BMP systems in gonadotrope cells. I have couple of questions about this study.

Major;

1. It is not quite clear why the authors focused on BMP, orexin, and circadian rhythm in gonadotropin synthesis. The authors should describe detailed rational in introduction.

2. The authors showed that enhanced expression of gonadotropins (LHb/FSHb) by orexin and/or GnHR stimulation was reversed with combination of orexin and GnHR. What is the mechanism underlying the reverse effect? How can inhibitory effect of orexin on GnRHR expression/enhanced effect of GnRH on orexin receptors (Fig1B) explain this?

3. Although the previous study of this group showed that clock genes (Bmal1 and Clock) were involved in of GnRH-mediated induction of LHb (ref# 14), it does not necessarily indicate that Bmal1/Clock mediate orexin-induced expression of gonadotropins. Are the clock genes involved in orexin-induced expression of gonadotropins? Does knockdown of clock genes by RNAi block induction of gonadotropin expression by orexin?

4. I do not find evidence indicating that BMPs have a positive effect on expression of gonadotropins (Fig 3). Also, in the author’s previous study, BMPs reversed or inhibited GnRH-induced expression of LHb (ref# 14). What date shows that BMP positively regulates gonadotropin expression?  

5. Regarding Fig 2B, it is not clear whether phosphorylation of smads were augmented by BMPs. The authors should show ratio of phosphor-smad/total smad for each smad proteins, otherwise it is difficult to interpret the results correctly.

Minor;

There is a typo in line 24 (OXR2 should be OX2R).

Author Response

REFEREE 1:
Comments and Suggestions for Authors
In this study, Soejima et al. studied the role of orexins, important neuropeptides involved in wakefulness, feeding behavior, emotion, and autonomic function in regulation of gonadotropin expression. Using mouse gonadotrope LbT2 cells, the authors examined the expression of gonadotropins (LHb and FSHb), GnRH-receptor, orexin receptors (OX1R and OX2R) and clock genes (Bmal1 and Clock) by stimulation with orexin and/or GnRH. Then, the authors examined expression of orexin receptors and clock genes by BMP stimulation. The authors also examined effect of orexin stimulation on BMP-induced phosphorylation of smad proteins, and expression of BMP receptors by orexin stimulation. The authors concluded mutual interaction of orexin, GnHR, and BMP systems in gonadotrope cells. I have couple of questions about this study.

Answer:  Thank you very much for your favorable review.  We revised our manuscript according to your constructive comments.  Our responses to the comments are shown in the following section.  We sincerely appreciate all of your constructive suggestions regarding our paper.

Major;
1. It is not quite clear why the authors focused on BMP, orexin, and circadian rhythm in gonadotropin synthesis. The authors should describe detailed rational in introduction.

Answer:  We agree with the comment.  We added a more detailed background of our research in the revised introduction section by adding a section regarding the circadian clock system.  We previously reported that orexins and the BMP system collaboratively modulate hormonal secretion in various endocrine organs (Otsuka. Endocr. J. 2010).  Orexin A enhanced FSH-induced progesterone production by downregulating BMP signaling in rat granulosa cells (Fujita, et al. J. Steroid Biochem. Mol. Biol. 2018).  Orexin A inhibited prolactin production by suppressing endogenous BMP activity in rat lactotrope cells (Fujisawa, et al. Peptides 2019) and enhanced POMC transcription by upregulating CRH-receptor signaling and by downregulating BMP signaling in mouse corticotrope cells (Fujisawa, et al. IJMS. 2021).  Thus, we focused on the functional interrelationship among orexin, BMP system and clock genes, targeting gonadotrope cells, in the present study.  Thank you for your opinion.

2. The authors showed that enhanced expression of gonadotropins (LHb/FSHb) by orexin and/or GnHR stimulation was reversed with combination of orexin and GnHR. What is the mechanism underlying the reverse effect? How can inhibitory effect of orexin on GnRHR expression/enhanced effect of GnRH on orexin receptors (Fig1B) explain this?

Answer:  We appreciate your comment.  Certainly, it was notable that enhanced expression of gonadotropins by orexin stimulation was reversed in the presence of GnRH treatments.  The molecular mechanism of dual effects of orexin was quite interesting in the aspect of the interaction with GnRH in this study.  As we addressed in the discussion section, it can be implied that orexin has modulatory effects on gonadotropin expression by downregulating GnRH responsiveness, leading to a feedback regulation of gonadotropin expression stimulated by GnRH.  On the other hand, GnRH directly stimulates gonadotropin expression by itself and also by enhancing the expression of orexin type 1 and type 2 receptors.  We added these contexts to the revised discussion.  Thank you for your comment.

3. Although the previous study of this group showed that clock genes (Bmal1 and Clock) were involved in of GnRH-mediated induction of LHb (ref# 14), it does not necessarily indicate that Bmal1/Clock mediate orexin-induced expression of gonadotropins. Are the clock genes involved in orexin-induced expression of gonadotropins? Does knockdown of clock genes by RNAi block induction of gonadotropin expression by orexin?

Answer:  As the referee commented, in the present study, we did not examine whether clock genes are directly involved in orexin-mediated gonadotropin expression by RNAi methods.  However, in our previous study (Soejima, et al. IJMS. 2021), we have already performed siRNA methods for knockdown of Bmal1 and Clock in mouse gonadotrope cells, and it was revealed that Bmal1 and Clock were functionally involved in the regulation of LHß expression, regardless of orexin effects, in the presence of GnRH.  In the present study, the experimental results showing that stimulation with orexin A, GnRH and BMPs enhanced clock gene expression and that clock genes mediated GnRH-induced gonadotropin expression indicated that clock genes also play a regulatory role in gonadotropin expression modulated by orexin.  Thank you for your constructive comments.

4. I do not find evidence indicating that BMPs have a positive effect on expression of gonadotropins (Fig 3). Also, in the author’s previous study, BMPs reversed or inhibited GnRH-induced expression of LHb (ref# 14). What date shows that BMP positively regulates gonadotropin expression?  

Answer:  We agree with the comments.  We previously demonstrated that BMPs positively regulate gonadotropin expression.  BMP-6 (1 to 10 ng/ml) and BMP-7 (1 to 10 ng/ml) treatment for 24 h enhanced transcriptional activities of gonadotropins and BMP-15 (1 to 100 ng/ml) treatment for 24 h also enhanced FSH transcriptional activity in LßT2 cells (Otsuka, et al. Endocrinology 2002).  We added these findings to the revised introduction section.  Thank you for your comments.

5. Regarding Fig 2B, it is not clear whether phosphorylation of smads were augmented by BMPs. The authors should show ratio of phosphor-smad/total smad for each smad proteins, otherwise it is difficult to interpret the results correctly.

Answer:  We appreciate the comments.  It was revealed that the density of phosphorylated-Smad1/5/9 was significantly increased by orexin A treatment induced by stimulation with BMPs.  On the contrary, that of total Smad1 was not significantly affected by the same treatment.  These findings were carefully confirmed by at least three independent experiments and this method was found to be more effective in order to compare the detailed changes in Smad1/5/9 phosphorylation caused by orexin treatment.

Minor;
There is a typo in line 24 (OXR2 should be OX2R).

Answer:  We appreciate the comment.  We corrected the mistake.

Thank you very much for the constructive comments.

Reviewer 2 Report

Soejima et al. examined the interactive effect of orexin, GnRH, and BMP on the gonadotropin expression in LßT2 cells, and found that there exists interaction among these, and the interactive regulation of Gn expression was possibly mediated by clock genes, such Bmal 1 and Clock.

The experimental results look to be straightforward, and the conclusions seem to sound. However, there are several concerns about the protocols of the experiment.

Lines 83-84: Why did the authors decide to incubate cells with orexin, GnRH, or BPTs for 24 h in the first experiment?

Do LßT2 cells show internal oscillation of clock genes? 

If LßT2 cells show internal oscillation of clock genes, the effect of orexin, GnRH, and BPTs could be time (circadian time) dependent, but the authors examined the impact on only one timepoint. Please explain briefly why the authors employ this protocol (24 exposure).

Also, it should be shown what time (circadian time) the authors started to apply these.

Lines 107-110: Why did the stimulation with BMP for 1 hour in this experiment?

Author Response

REFEREE 2:
Comments and Suggestions for Authors
Soejima et al. examined the interactive effect of orexin, GnRH, and BMP on the gonadotropin expression in LßT2 cells, and found that there exists interaction among these, and the interactive regulation of Gn expression was possibly mediated by clock genes, such Bmal 1 and Clock.
 The experimental results look to be straightforward, and the conclusions seem to sound. However, there are several concerns about the protocols of the experiment.

Answer:  Thank you very much for your favorable comments.  According to your comments, we revised our manuscript.  Thank you for your opinion.

Lines 83-84: Why did the authors decide to incubate cells with orexin, GnRH, or BMPs for 24 h in the first experiment?

Answer:  We appreciate your comment.  In our previous study including a time-course study (Soejima, et al. IJMS. 2021), it was found that the expression levels of LHß were markedly elevated after GnRH stimulation for 24-h incubation rather than the time course for 1 h, 3 h, 6 h or 12 h.  Therefore, we selected the incubation time of 24 h.  We added these findings to the revised methods section.  Thank you for your comments. 

Do LßT2 cells show internal oscillation of clock genes? 

Answer:  Yes.  In our earlier study (Soejima, et al. IJMS. 2021), it was revealed that the expression levels of clock genes, including Clock, Bmal1, Per2 and Cry1, exhibited oscillational changes regardless of the induction of GnRH in LßT2 cells.  In the GnRH-free condition, expression of clock genes was elevated at 1-6 h and was decreased at 24 h after the serum-free exposure.  In the presence of GnRH, expression of clock genes was upregulated compared with that in control groups at 24 h.  We added this information to the revised discussion section.

If LßT2 cells show internal oscillation of clock genes, the effect of orexin, GnRH, and BMPs could be time (circadian time) dependent, but the authors examined the impact on only one timepoint. Please explain briefly why the authors employ this protocol (24 exposure).

Answer:  As we addressed in the former question, we chose the timepoint of 24 h in the present study since the expression levels of LHß were markedly elevated after GnRH stimulation for 24-h incubation rather than the time course for 1 h, 3 h, 6 h or 12 h (Soejima, et al. IJMS. 2021).  We added these findings to the revised methods section.  Data at different timepoints will be examined in our future research.  Thank you for your constructive comments.

Also, it should be shown what time (circadian time) the authors started to apply these.

Answer:  Thank you for your comment.  We synchronized the circadian rhythm of LßT2 cells by exposing cells to serum-free conditions for 3 h as performed in the previous study.  We added this issue to the revised methods section.

Lines 107-110: Why did the stimulation with BMP for 1 hour in this experiment?

Answer:  Thank you for your comment.  Based on our previous experiments (including Takeda, et al. Mol. Cell. Endocrinol. 2012, and Toma, et al. Peptides 2016), we found that 1-h stimulation with BMP ligands was appropriate to reproduce the detection of effective phosphorylation of Smads in mouse gonadotrope LßT2 cells.  We added the description in the revised methods section.

Thank you very much for the constructive comments.

Round 2

Reviewer 1 Report

The authors answered all my questions. I'm satisfied.

Author Response

REFEREE 1:
Comments and Suggestions for Authors
The authors answered all my questions. I'm satisfied.

Answer:  Thank you for your review.

Reviewer 2 Report

The revised manuscript is improved substantially, but there is a point that I am not satisfied with the response of the author's response and revision.

+++

Also, it should be shown what time (circadian time) the authors started to apply these.

Answer:  Thank you for your comment.  We synchronized the circadian rhythm of LßT2 cells by exposing cells to serum-free conditions for 3 h as performed in the previous study.  We added this issue to the revised methods section.

+++

I asked what circadian time the author examined the effect of orexin, GnRH, and BMPs on Gns expression. It may be the question was not clear, and you may not have understood the meaning of my question. I asked what phase of the circadian phase you examined the effect of orexin, GnRH, and BMPs on Gns expression. Did you examine during the phase with higher Bmal1/Clock, or lower Bmal1/Clock? Why did you determine the phase with higher or lower Bmal1/Clock to examine the effect of orexin, GnRH, and BMPs?

I think it is an important issue to elucidate the role of Bmal1/Clock.

Author Response

REFEREE 2:
Comments and Suggestions for Authors
The revised manuscript is improved substantially, but there is a point that I am not satisfied with the response of the author's response and revision.

Also, it should be shown what time (circadian time) the authors started to apply these.  I asked what circadian time the author examined the effect of orexin, GnRH, and BMPs on Gns expression. It may be the question was not clear, and you may not have understood the meaning of my question. Q1) I asked what phase of the circadian phase you examined the effect of orexin, GnRH, and BMPs on Gns expression. Q2) Did you examine during the phase with higher Bmal1/Clock, or lower Bmal1/Clock? Q3) Why did you determine the phase with higher or lower Bmal1/Clock to examine the effect of orexin, GnRH, and BMPs?  I think it is an important issue to elucidate the role of Bmal1/Clock.

Answer:  Thank you for your constructive comment again.  Based on our previous study (Soejima, et al. IJMS. 2021), we found that Bmal1 and Clock mRNA levels were elevated at 1 to 12 h (the early phase) and decreased at 12 to 24 h (the late phase) after the serum-free exposure in the absence of GnRH by LbetaT2 cells.  However, in the presence of GnRH, the expression levels of Bmal1 and Clock were lowered in the early phase, and then in the late phase, the Bmal1 and Clock expression was highly increased compared to GnRH-free condition.  To characterize the expressional changes of LHbeta, Bmal1 and Clock mRNA in the presence / absence of GnRH stimuli, we here attached Figure A showing the serial changes of these expressions.  As seen in this figure below, at the 24-h time point of our culture system of LbetaT2 cells, Bmal1 and Clock gene expression are attained to be the highest by GnRH treatment.

Therefore, our answer to Q1) is that we determined the timepoint at 24 h when GnRH effects are highly affected to upregulate Bmal1 and Clock gene expression; and our answer to Q2) is that: Bmal1 and Clock mRNA expression is effectively elevated by GnRH stimulation during the phase around 24 h.  And the answer to Q3) is that: since the time point of 24-stimulation of GnRH was stable to demonstrate the effects of GnRH on gonadotropin and the related-clock gene expressions in our system, we utilize the “24-h time point” to clarify the effects of orexin and BMPs on Bmal and Clock gene expression in the present study.

As a result, as we presented the data in Fig. 1C and 2A, showing that orexin A and BMPs upregulated clock expression after the time point of 24-h treatment.  Considering the present results, it was concluded that orexin A and BMPs as well as GnRH can upregulate Bmal, Clock and gonadotropin expression, suggesting that the effects of orexin, BMPs and GnRH seem to be functionally linked to the clock gene expression in the process of gonadotropin induction.

Thus, we examined the phase showing effectively higher expressions of gonadotropin and clock genes by GnRH stimulation in our culture system, which was the “24-h time point” after the GnRH stimulation.  In the future work, we would like to examine clock gene expression at the different time points.  We added this information to the revised methods and discussion section.

Thank you very much for the constructive comments again.

Round 3

Reviewer 2 Report

The authors have responded appropriately to my concern appropriately. I have no more comments.